# The relationship between internet gaming disorder, sleeping quality, self-compassion, physical activity participation and psychological distress: A path analysis

internet gaming disorder; self-compassion; sleep; physical activity

**Corresponding authors:**
Ming Yu Claudia Wong and Hong Wang Fung;
Emails: cmywong@eduhk.hk;
andyhwfung@gmail.com

Ming Yu Claudia Wong[1] , Guangzhe Frank Yuan[2,3], Caimeng Liu[4],
Stanley Kam Ki Lam[5] and Hong Wang Fung[6]

[1]Department of Health and Physical Education, The Hong Kong Education University, New Territories, Hong Kong; [2]South Carolina SmartState Center for Healthcare Quality, Arnold School of Public Health, University of South Carolina, Columbia, SC, USA; [3]Department of Health Promotion, Education, and Behavior, Arnold School of Public Health, University of South Carolina, Columbia, SC, USA; [4]College of Teacher Education, Institute of Education Science, Leshan Normal University, Leshan, China; [5]The Nethersole School of Nursing, Faculty of Medicine, The Chinese University of Hong Kong, New Territories, Hong Kong and [6]Department of Social Work, Hong Kong Baptist University, Hong Kong, School of Nursing and Health Studies, Hong Kong Metropolitan University, Hong Kong

## Abstract

The classification of internet gaming disorder (IGD) as a mental condition for further study in 2013 marked the emerging recognition of potential mental health issues associated with internet and gaming addiction. The COVID-19 pandemic and the rapid growth of gaming technology have combined to increase internet gaming, resulting in unhealthy lifestyle behaviors, poor sleep quality and psychological distress. Identifying the complex interplay between internet problem use, sleep disorders and psychological distress is crucial. However, it remains unclear how physical activity and self-compassion could improve sleep quality when individuals experience IGD symptoms. The current study, therefore, examined the relationships between IGD, sleep quality, self-compassion, physical activity and psychological distress using a path analysis approach. The study, targeting young adults ($N$ = 283), found that physical activity played a significant role in connecting the variables and supporting the overall fit of the model. The results suggest that interventions targeting individuals with IGD should focus on promoting physical activity participation and developing self-compassion. Future research should continue to investigate the effectiveness of clinical interventions that incorporate self-compassion and physical activity counseling for individuals with IGD.

## Impact statement:

This study sheds light on the important relationships between internet gaming disorder, sleep quality, self-compassion, physical activity and psychological distress. It finds that internet gaming disorder is negatively associated with sleep quality and positively linked to depression, stress and psychological distress. However, cultivating self-compassion and engaging in physical exercise can help reduce the negative impacts of internet gaming disorder. Specifically, self-compassion and physical activity were found to benefit psychological health by better equipping individuals to cope with stressors from internet gaming addiction. These findings point to promising intervention strategies and support the potential role of lifestyle factors in mitigating the risks from internet gaming disorder. The study suggests combining self-compassion education with active video gaming and a gradual transition to other physical activities, alongside motivational counseling. Such a multipronged behavioral modification approach integrating physical stimulation with psychoeducation is likely to yield more sustained effects than attempts to purely control gaming behaviors. It represents a more acceptable and feasible way of implementing interventions in real-world clinical settings. Overall, the study provides insights toward establishing evidence-based strategies for addressing the significant and growing public health challenge of problematic internet use, especially among youth.

### Key practitioner messages:
- Among the relationship between internet gaming disorder (IGD), sleep quality, self-compassion, physical activity and psychological distress in young adults, physical activity plays a significant role in connecting the variables and supporting the overall fit of the model. It is suggested that interventions targeting individuals with IGD should focus on promoting physical activity participation.



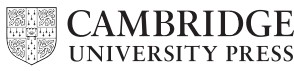

- As a result of the good-fit model, emphasis has also been placed on the importance of promoting physical activity and self-compassion as ways to improve sleep quality and reduce psychological distress in individuals with IGD.
- Future practical and research activities should continue to investigate the effectiveness of clinical interventions that incorporate self-compassion and physical activity counseling for individuals with IGD symptoms.

## Background

In recent years, internet gaming disorder (IGD) has emerged as a significant concern. Statistics showed that 1.2%–8.5% of adolescents and adults worldwide appear to be diagnosed with IGD symptoms (Kuss et al., 2017). This disorder denotes an alarming gaming behavior pattern characterized by excessive usage of internet games that has led to substantial impairment and distress. IGD was not yet defined as a diagnostic mental disorder when the DSM-IV was published in 1994. Yet, since the emerging recognition of the potential mental health issues due to internet and gaming addiction, IGD was classified as a mental condition for further study in 2013. In the DSM-IV session, it was explained that people who are preoccupied with gaming, have withdrawal symptoms when playing is not possible (*i.e.,* emotional instability, anxiety), lose interest in previously enjoyed activities due to gaming, or even give up other compulsory activities (*i.e.,* school, work, social networking), along with six other networking-related issues, will be diagnosed as IGD (Kardefelt-Winther et al., 2017; Kuss et al., 2017). For this reason, the current research aims to investigate the relationship between IGD, withdrawal symptoms (mainly psychological distress) and sleep quality; additionally uncovers how positive psychological concepts contribute to this relationship. As the possibility of declining interest in activities while addicted to internet gaming, the importance of physical activity is emphasized, which aims to advance the literature on how leading a healthy lifestyle can lessen the negative consequences of IGD, thus providing the field with quantitative evidence in supporting positive psychological and healthy lifestyle practices in reducing video gaming, internet addiction and any other detrimental consequences of excessive screen time.

### IGD and sleeping quality

The rapid growth of gaming technology and gaming products, as well as the social distancing measures due to the COVID-19 pandemic, have led to an increase in internet gaming, exacerbating the negative consequences associated with this behavior, including psychological distress (*i.e.,* anxiety, depression), poor sleep quality (Tsui and Cheng, 2021), impaired social functioning, poor academic performance (Kuss et al., 2017) and unhealthy lifestyle behaviors (Wong et al., 2020). The relationship between problematic internet use and psychological distress was well-established in the field, indicating that internet games can cause stress, anxiety

and depression in a direct way, while aligning with IGD symptoms (Lam, 2014). In addition to anxiety and depression, the systematic review conducted by Lam in 2014 established a connection between problematic internet use and sleep disorders. Lam's (2014) systematic review summarized that the association between problematic internet use and depression might involve different potential causal and cross-sectional relationships, such as sleep problems and bio-psychosocial distress (*i.e.,* attention-deficit/hyperactivity disorder, sensation-seeking, inconsistent cognition of "self," peer problems, aggressive and severe rule-breaking behaviors) (Sugaya et al., 2019). Research has also revealed the significant mediating effect of sleep quality on IGD and subjective well-being, especially when considering the influence of self-control among adolescents (Li et al., 2022). On the other hand, the research found that psychological distress acted as a mediator between IGD and sleeping problems, as well as between IGD and quality of life (Fazeli et al., 2020). This underscores the importance of undertaking research to gain a comprehension of the intricate interplay between problem internet use, sleep disorders and various psychological distresses. Based on the above literature review, research expected that IGD will lead to stress and depression thus further affecting sleeping quality, while IGD was also tested to be associated with sleeping quality (Figure 1).

### Self-compassion and healthy lifestyle behaviors as potential mediator(s)

Self-compassion has gained considerable attention in mental health research over the past decade. Self-compassion is described as a way of treating oneself with kindness, responding to deficiencies and sufferings with emotional equilibrium without overidentifying or isolating them. Significant research studies have revealed self-compassion as a factor in preventing or reducing vulnerability caused by mental health problems. Self-compassion was mostly associated with higher resilience and less psychological distress symptoms like depression, stress, anxiety and physical and mental exhaustion (Neff, 2011; Tendhar et al., 2022). In addition, self-compassionate individuals were less likely to experience depressive symptoms and reported more positive emotions, thus resulting in better mental health outcomes and a higher level of life satisfaction (MacBeth and Gumley, 2012; Neff and Germer, 2013). In contrast, less self-compassionate individuals would engage in frequent self-critique, thus increasing the risk of being associated with different psychological and emotional distress (Gilbert and Procter,

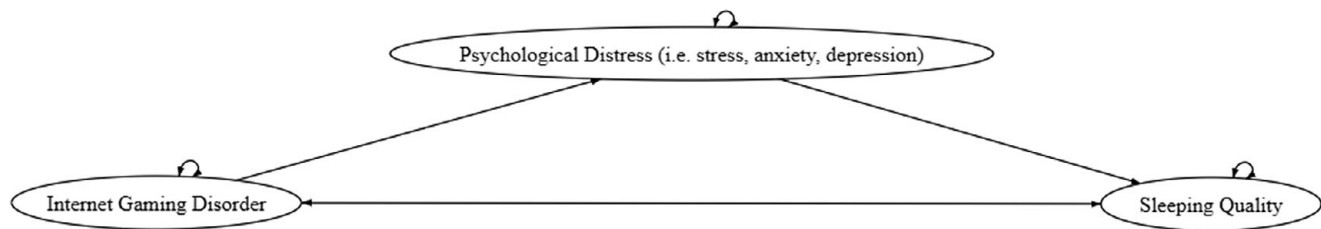

**Figure 1.** Reviewed relationship between internet gaming disorder, psychological distress and sleeping quality.

2006). Furthermore, research suggested a significant interrelationship between self-compassion, depression, stress (including interpersonal stress) and sleep. Multiple systematic reviews and meta-analysis studies documented the substantial association between self-compassion and better sleep quality, as well as significantly predicting other health-related domains, such as better physical health and merged health behaviors (; Butz and Stahlberg, 2020; Brown et al., 2021; Cha et al., 2022). Further research revealed that both healthy students and depressive disorder patients showed a significant decrease in ruminative thoughts and improved sleeping quality after a week of daily self-compassion intervention practice (Butz and Stahlberg, 2018). Model analysis research studies also provided insights into the mechanisms underlying self-compassion and sleeping quality. They supported the direct and indirect effects caused by predisposing and precipitating factors of depression and anxiety, including interpersonal problems, perceived stress due to current life-challenging events and rumination (Bian et al., 2022; Rakhimov et al., 2022). Therefore, highlighting the importance of self-compassion on nonjudgmental and emotional equilibrium internal support; along with facilitating the development of resilience and emotional coping skills, self-compassion is a preventive measure for mental health issues, such as psychological distress, as well as other mental disorders, such as IGD.

As well as practicing self-compassion, healthy behaviors like participating in physical exercise, in which "exercise is medicine" (Thompson et al., 2020), were also considered preventive measures for mental issues and sleep problems. Reviews summarized the positive association between physical activity and sleep quality, as well as the effectiveness of depressive symptoms on the level of physical activity and sleep (Sampasa-Kanyinga et al., 2020; Atoui et al., 2021). Notwithstanding, the significant relationship between self-compassion, physical activity and mental well-being was also indicated by the author and other researchers (Biber and Ellis, 2019; Wong et al., 2021a,b, 2023). Self-compassion was shown to be highly correlated with higher level engagement in actual physical activity and exercise, as well as higher physical activity participation intentions (Biber and Ellis, 2019; Hallion et al., 2019).

Despite there being a few research studies indicating the association between regular physical activity and a lower risk of internet

gaming addiction (Hazar and Hazar, 2018; Gülü et al., 2023), the effect mechanism of physical activity, so as self-compassion, on enhancing individuals' sleeping quality due to the influences of IGD symptoms, was not yet investigated.

### Research purpose and hypothesis

In this study, we hypothesized that the relationship between IGD and sleep quality would be mediated by interpersonal stress and depression symptoms. Furthermore, we predicted that individuals with high levels of self-compassion (mindfulness and self-kindness) and regular exercise behavior are likely to experience a positive impact on sleep quality, particularly concerning IGD and psychological distress. Conversely, we expected that individuals with prominent levels of self-judgment would harm the relationship. It is important to note that the measurement model presented in the path model will be varied based on the confirmatory factor analysis (CFA) outcome. To test these hypotheses, the study conducted a path analysis to investigate the following hypothesized path model (Figure 2).

The current study, therefore, aims to investigate the relationships between IGD, sleep quality, self-compassion, physical activity and psychological distress using a path analysis approach. By examining the underlying mechanisms and potential pathways involved, this study seeks to contribute to a better understanding of the complex relationships between IGD, sleep quality, self-compassion, physical activity and psychological distress and provide insights into the development of effective interventions and prevention strategies.

### Methods

#### Participants

We examined the relationships among the study variables using data from an international survey project that investigated psychosocial experiences and health problems among young people. From February to March, 2023, potential participants were recruited using online methods (through Facebook and Instagram), which have been commonly used in health and social services and research (Chan et al., 2017; Fung et al., 2020; Fung et al., 2022). The inclusion criteria were as follows: 1) being aged 18–24; 2) could read and write

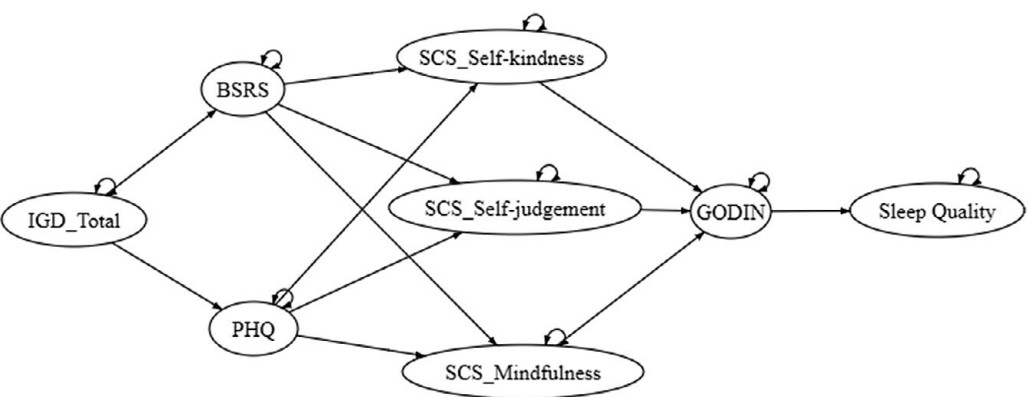

**Figure 2.** Hypothesized path model.
*Note*: BSRS, Bergen Social Relationships Scale; GODIN, Godin-Shephard Leisure-Time Physical Activity Questionnaire; IGD_Total, Internet Gaming Disorder Symptoms Total; PHQ, Patient Health Questionnaire; SCS_selfkindness, Self-Compassion (Self-Kindness); SCS_mindfulness, Self-Compassion (Mindfulness); SCS_judgment, Self-Compassion (Self-Judgment).

English and 3) agree to provide informed consent and participate in the online survey study. Participants with a clinical diagnosis of a reading disorder, dementia or intellectual disabilities were excluded. The methodology, including part of the baseline data, has been published elsewhere (Fung et al., 2023).

## Measures

The following validated standardized assessment tools were included in the survey.

The IGD Scale–Short-Form (IGDS9-SF) has nine items and is a self-report screening tool for assessing the nine DSM-5 criteria for IGD (Pontes and Griffiths, 2015). A systematic review reported that the IGDS9-SF had adequate internal consistency and excellent validity (Poon et al., 2021). The total score may range from 9 to 45; higher scores indicate higher levels of IGD symptoms. The IGDS9-SF had good internal consistency ($\alpha$ = .881) in our sample.

The Bergen Social Relationships Scale (BSRS) has six items and is a brief measure assessing the level of interpersonal stress (Bancila and Mittelmark, 2009). The BSRS was reported to have satisfactory internal consistency ($\alpha$ = .73) and a stable factor structure across cultural contexts (Bancila and Mittelmark, 2009). The total score may range from 6 to 24; higher scores indicate more interpersonal stress. The BSRS had acceptable internal consistency ($\alpha$ = .754) in our sample.

The Patient Health Questionnaire-Brief (PHQ-B) is a shortened version of the 9-item PHQ-9, which is a widely used screening measure for depressive symptoms (Kroenke et al., 2001; Kung et al., 2013). Manea et al. (2016) have demonstrated that a shortened version of the PHQ also has good psychometric properties. Informed by the study conducted by Manea et al. (2016), we used Items 1 and 2 of the PHQ-9 to screen for depressive symptoms. In addition to these two items, we also included Item 9 of the PHQ-9 for two major reasons. First, suicide is a major feature and considerable outcome associated with depression. A number of studies have shown that the PHQ-9 Item 9 could effectively and independently predict depression-related suicidal risk (*e.g.*, Simon et al., 2013; Chung et al., 2023). Second, in an international sample (N = 468), we found that, compared with the PHQ-2 (Items 1 and 2 only), the PHQ-B that included Items 1, 2 and 9 had a stronger correlation with the full PHQ-9 total score (r = .847 vs. r = .881, p < .001) (unpublished data from Fung et al., 2023). Therefore, including Item 9 in addition to the PHQ-2 based on Manea et al. (2016) could better capture the diverse dimensions of depressive symptoms. Also, since Item 9 is originally included in the PHQ-9, adding this item would only increase the validity of the screening measure. Therefore, in this study, we included Items 1, 2 and 9 of the PHQ-9 in this study. The PHQ-B had good internal consistency ($\alpha$ = .815) in this sample.

The Self-Compassion Scale-Short Form (SCS-SF) is a shortened version of the original 26-item SCS, which is a reliable and valid measure of self-compassion, with six established factors, namely self-kindness, common humanity, mindfulness, self-judgment, isolation and overidentification (Neff, 2016; Neff et al., 2019). Although the SCS-SF only had 10 items, it was also found to have sound psychometric properties (Raes et al., 2011). In this study, only the Self-kindness, Self-judgment and Mindfulness subscales were used. Higher scores indicated higher levels of the specific form of self-compassion.

The Godin-Shephard Leisure-Time Physical Activity Questionnaire is a 4-item self-report measure of physical exercise frequency within the past 7 days, with established reliability and validity (Godin, 2011). This measure has been commonly used in health research (Amireault et al., 2015). It assesses three types of physical exercise, including strenuous, moderate and light exercise. The total leisure activity score = (strenuous exercise × 9) + (moderate exercise × 5) + (light exercise × 3). Higher scores indicate more physical exercise in the past week.

The single-item Sleeping Quality Scale, which asks "During the past 7 days, how would you rate your sleep quality overall?" (Terrible = 0; Poor = 1 to 3; Fair = 4 to 6; Good = 7 to 9; Excellent = 10). This single item was found to have good reliability, validity and responsiveness to assess sleep quality (Cappelleri et al., 2009).

## Data analysis

The SPSS 28 software was used to record the demographic information of the participants, including age and gender. Descriptive statistics were used to synthesize data on physical activity, depression symptoms, interpersonal distress, IGD level and self-compassion. Basic associations between the variables were examined using the Pearson correlation. The CFA of the measurement models and the path analysis models were constructed and tested using path analysis in LISREL 12, with the full-information maximum likelihood estimator. The goodness of fit of the models was evaluated based on several criteria, including a chi-square ranging from 2 to 5, a comparative fit index (CFI) and non-normed fit index rating of 0.90 or above, a standardized root mean square residual (SRMR) value of 0.08 or below, and a root mean square error of approximation (RMSEA) value of 0.08 or below, with a 90% confidence interval that holds within this value considered indicative of a good fit (Jaccard and Wan, 1996; Hu and Bentler, 1999). The data analysis is expected to explore the relationships between the variables and identify potential mediating factors.

In addition to the rule-of-thumb for path analysis, a sample size sensitivity analysis was performed to determine the significance of the sample size in identifying significant paths. The analysis was conducted using the *lavaan* and *simsem* packages in R Studio, which utilize Monte Carlo simulations in structural equation modeling. According to Muthén and Muthén (2002), sample size power is determined by the standard error biases and the coverage values of the parameters. A sufficient power close to 0.8 was achieved when the standard error was within 10% of any parameter and 5% of the parameter, with coverage remaining between 0.91 and 0.98. Moreover, the standard alpha threshold of 0.05 was applied. The lowest recommended power threshold of 0.8 was chosen for power to minimize Type II error.

## Results

There were 283 participants (91.2% of females) of aged 18–24 involved in the current survey. Tables 1 and 2 present the demographic, descriptive statistics and the correlation matrix of the measurement variables.

## Confirmatory factor analysis

Table 3 presents all the CFA results of the measurement models before conducting the path analysis. The table also indicates the Cronbach's alpha and Omega values of each measurement model. Most of the measurement models showed Cronbach's alpha and

**Table 1.** Demographic information and descriptive statistics

| Variable | Mean | SD |
|---|---|---|
| Age | 20.04 | 1.88 |
| Internet Gaming Disorder Symptoms Total | 14.02 | 6.22 |
| Self–Compassion (Self–Kindness) | 5.05/10 | 1.96 |
| Self–Compassion (Self–Judgment) | 7.49/10 | 1.86 |
| Self–Compassion (Self–Mindfulness) | 6.09/10 | 1.90 |
| Patient Health Questionnaire (PHQ) – Depressive symptoms | 5.34 | 2.66 |
| Bergen Social Relationships Scale (BSRS) – Interpersonal Stress | 15.35 | 4.33 |
| Godin–Shephard Leisure–Time Physical Activity Questionnaire | 39.89 | 36.22 |
| Sleeping Quality – "During the past 7 days, how would you rate your sleep quality overall?" | 4.77 | 2.48 |

**Table 4.** Standardized direct, indirect and total effect of internet gaming disorder on sleep quality

| Pathways | Direct effect | Indirect effect | Total effect |
|---|---|---|---|
| Internet Gaming Disorder → BSRS&PHQ → Self–compassion | – | −0.039** | −0.039** |
| Internet Gaming Disorder → BSRS&PHQ → Self–compassion →Physical Activity | – | −0.019** | −0.019** |
| Internet Gaming Disorder → BSRS&PHQ → Self–compassion →Physical Activity→ Sleep Quality | – | −0.0006** | – |
| Internet Gaming Disorder→ Sleep Quality | – | – | −0.000032** |

$**p < 0.01$,
$***p < 0.001$.

Omega values above 0.7, indicating a satisfactory internal consistency. The models also demonstrate a satisfactory goodness of fit, thus enabling investigation using path analysis. It is worth noting that, as self-judgment is an opposite latent variable of self-compassion, it did not show a goodness of fit in this sample population despite after reverse coding and summed with the total score of self-kindness and mindfulness; hence, only scoring from self-kindness and mindfulness were used in the current analysis. Nevertheless, the correlation matrix in Table 2 demonstrated the significant negative association between self-judgment and sleeping quality.

## Path analysis

The total scores of the measurement scales were computed to perform path analysis as observed variables. Based on the hypothesized model, the path analysis results demonstrated acceptable goodness of fit, with CFI = 0.90, TLI = 0.87, SRMR = 0.078 and RMSEA = 0.10 (90% CI =0.063–0.14), with the error covariance were set up between PHQ and BSRS and PHQ and sleeping quality. However, nonsignificant paths were shown between BSRS and self-compassion, and between self-compassion, physical activity and sleeping quality. Despite removing the physical activity variable resulting in a significant pathway between self-compassion and

**Table 2.** Correlation matrix of the variables

| Variable | GODIN | SCS_selfkind | SCS_judg | SCS_mind | IGD_total | PHQ | BSRS | Sleep quality |
|---|---|---|---|---|---|---|---|---|
| GODIN | 1.00 | | | | | | | |
| SCS_selfkind | 0.03 | 1.00 | | | | | | |
| SCS_judg | −0.01 | −0.50** | 1.00 | | | | | |
| SCS_mind | 0.06 | 0.59** | −0.19** | 1.00 | | | | |
| IGD_total | −0.09 | −0.08 | 0.11 | −0.06 | 1.00 | | | |
| PHQ | −0.04 | −0.48** | 0.28** | −0.32** | 0.16** | 1.00 | | |
| BSRS | −0.02 | −0.26** | 0.29** | −0.12 | 0.24** | 0.40** | 1.00 | |
| Sleep quality | 0.03 | 0.18** | −0.15** | 0.17** | −0.08 | −0.33** | −0.29** | 1.00 |

*Note*: Significant correlations at the 0.01 level (two-tailed) are indicated by "**". Significant correlations at the 0.05 level (two-tailed) are indicated by "*".
BSRS, Bergen Social Relationships Scale; GSLTPAQ, Godin-Shephard Leisure-Time Physical Activity Questionnaire; IGD_Total, Internet Gaming Disorder Symptoms Total; PHQ, Patient Health Questionnaire; SCS_selfkind, Self-Compassion (Self-Kindness); SCS_mind, Self-Compassion (Mindfulness); SCS_judg, Self-Compassion (Self-Judgment).

**Table 3.** Measurement model goodness of fit index

| Measurement | Cronbach's alpha | Omega | Chi-square | CFI | TLI | SRMS | RMSEA |
|---|---|---|---|---|---|---|---|
| Bergen Social Relationships Scale (BSRS) | 0.75 | 0.76 | 17.73/9 = 1.9 | 0.97 | 0.95 | 0.047 | 0.068 (90% = 0.015–0.11) |
| Internet Gaming Disorder Symptoms Total (IGD) | 0.88 | 0.89 | 98.59/29 = 3.3 | 0.92 | 0.89 | 0.056 | 0.12(90% = 0.089–0.14) |
| Self–compassion (Mindfulness and Self–kindness) | 0.75 | 0.72 | 2.011/2 = 1.0 | 1.00 | 1.00 | 0.018 | 0.005(90% = 0.00–0.14) |
| Patient Health Questionnaire (PHQ) | 0.82 | 0.82 | 0 | 1.00 | 1.00 | 0 | 0.0 |
| Internet Gaming Disorder Symptoms Total (IGD) | 0.55 | 0.61 | 0 | 1.0 | 1.0 | 0 | 0.0 |

**Table 5.** Sample size sensitivity analysis

| Path | Average estimate | Average SE | Coverage |
|---|---|---|---|
| y ~ x1 | 0.03000071 | 0.0003487785 | 0.96 |
| y ~ x2 | 0.05000098 | 0.0004053609 | 0.92 |
| y ~ x3 | 0.00002960852 | 0.0004103136 | 0.93 |
| y ~ x4 | 0.2400204 | 0.0003888445 | 0.95 |
| y ~ x5 | 0.1399616 | 0.0003419795 | 0.95 |
| y ~ ~y | 0.00003157592 | 0.00000265447 | 0.94 |

### Power analysis

A power analysis was conducted using the "simsem" function in R Studio, and the resulting estimates were used to determine the appropriate sample size for the study (n = 283). The analysis of the estimated average parameters revealed a range of 0.00003–0.24, with an average standardized error range of 0.000002–0.0004 that is mostly within 10 % of the parameters except for the negative parameter. The coverage of the parameters ranged from 0.93 to 0.95, and the confidence intervals for the lower and upper bounds did not include zero, indicating statistical significance. These results (Table 5) demonstrate that the sample size used in the study is sufficiently powered and sensitive to detect meaningful effects.

sleep quality, the goodness of fit of the model was seriously reduced (*i.e.,* CFI = 0.48). This suggested that physical activity played a role in connecting the variables and supporting the model's overall fit. This also supports the role of physical activity in this mechanism; however, the representativeness of the sample might be questioned. By comparing the hypothesized model (Figure 2) and the observed path (Figure 3) analysis outcome, the two models matched to a larger extent, except for the use of the three-factor model of self-compassion in the data analysis due to the low goodness of fit of the measurement model. The path associations were well predicted as well, with IGD positively associated with BSRS and PHQ, then negatively associated with self-compassion, and leading to the change in sleep quality (Table 4).

Concerning the demographic variables as potential covariance, the invariance model analysis was conducted to indicate the existence of significant differences between the configural and scalar models. Gender, education level and family dependency were included in the invariance model analysis. The invariance model analysis indicated there were no significant differences between the different demographic variables, with p > 0.05 in the compare model fit outcomes. The invariance models kept the same CFI and TLI throughout, with the SRMS ranging from 0.028 to 0.042 and the RMSEA ranging from 0.001 to 0.073 across models and variables. It is, therefore, indicated that the model established equality across gender, education level and family dependency.

Overall, the path analysis model showed that IGD was positively associated with interpersonal stress and depression symptoms and that self-compassion and exercise behavior mediated the relationship between depression symptoms and sleep quality.

### Discussion

The current study investigated the relationship between IGD, sleeping quality, self-compassion, physical activity participation and psychological distress. The results of CFA showed that most of the measurement models had satisfactory internal consistency and goodness of fit, which enabled the researchers to investigate the hypothesized model using path analysis. However, the self-judgment measurement model did not show goodness of fit in this sample population, and only the scores from self-kindness and mindfulness were used in the analysis. The path analysis results demonstrated acceptable goodness of fit, with physical activity playing a role in connecting the variables and supporting the overall fit of the model. The mediation model was analyzed using the Andrew F. Hayes Process 4.1 Model 4, which indicated an overall significant regression model with IGD, psychological distress, self-compassion and physical activity as predictors of sleeping quality.

The results of the current study shed substantial insights into the mechanism of IGD and sleeping quality through self-compassion, physical activity participation and psychological distress. The path analysis model demonstrated acceptable goodness of fit. This suggested that IGD was positively associated with interpersonal stress and depression symptoms, and negatively associated with self-compassion and exercise behavior, which ultimately led to poor sleep quality. These results are consistent with previous research that shows a link between IGD and poor mental health outcomes, including depression and anxiety (Lam, 2014). In the literature review session of the current manuscript, it showed a few works

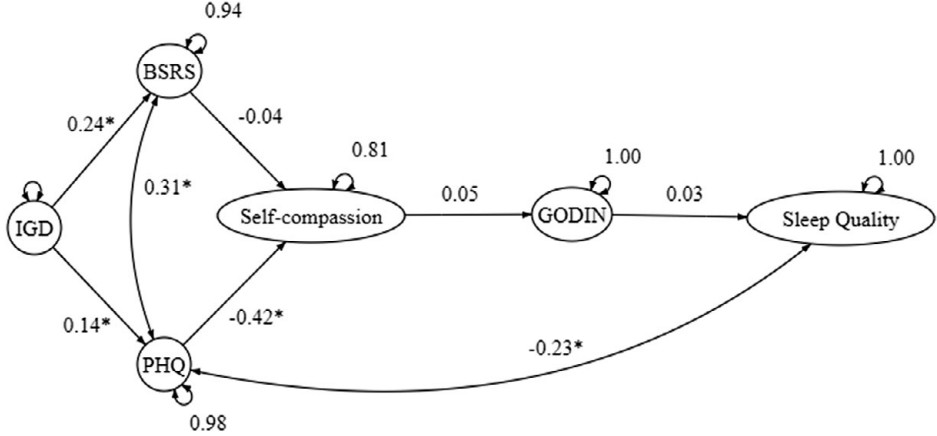

**Figure 3.** The path analysis model.
*Note*: Path coefficient in standardized β. BSRS, The Bergen Social Relationship Scale; GSLTPAQ, Godin-Shephard Leisure-Time Physical Activity Questionnaire; IGD_Total, Internet Gaming Disorder Symptoms Total; PHQ, Patient Health Questionnaire. *$p$ < 0.05.

of literature reviewing the effect of IGD on sleep alone, thus including reviews related to problematic internet use and addiction problems and citing their related effect on health-related consequences. Although the DSM-V recognizes IGD as a separate issue from other problematic internet use or digital addictions, research has shown that the dimensions of general problematic internet use are intricately linked to those of IGD, resulting in overlapping symptoms and behaviors. Furthermore, the s study demonstrated that both IGD (IGD) and general problematic internet usage share a substantial negative correlation with the quality of life among adolescents (Machimbarrena et al., 2022). The study identified a robust correlation between IGD and addiction to Facebook. Furthermore, the pathways' coefficient indicated that both disorders were equally linked to depression, anxiety and stress (Pontes, 2017). Based on this circumstance, the current path model appears highly generalizable to different problematic digital usage issues, such as problematic internet use and addiction. However, future research should focus on exploring the specific effects of IGD on health-related consequences and distinguishing those effects from other problematic digital usage issues.

Interestingly, the study found that physical activity participation played a significant role in connecting the variables and supporting the overall fit of the model. This finding further bolsters the evidence that physical activity fosters better sleep quality. Additionally, engaging in regular physical activity has been linked to improving sleep quality in the control of interpersonal stress and depression. Research studies have revealed adolescents with lower physical activity levels and higher sedentary behavior tended to be related to a higher risk of internet addiction (Han et al., 2021). Although it is recommended to encourage physical activity as a way to reduce internet addiction, some studies suggest that interventions focused on behavior modification to reduce sedentary behavior and increase physical activity may only yield a slight effect. Furthermore, the adherence effect after the intervention was not found to be significant according to research (Throwvala et al., 2021). As a result, research studies suggested that fundamental movement play showed greater improvement in reducing children's screen time compared to that of solely behavior modification education intervention (Salmon et al., 2005); furthermore, replacing non-active video gaming with active video gaming could be a first step toward reducing internet addiction before transitioning to physical activity. (Simons et al., 2015).

In addition, it supports the well-established role of self-compassion in this regard, as noted in previous literature review sessions. In essence, it provides evidence that self-compassion, entailing self-kindness and mindful self-awareness, functions as a critical protective factor against depression and anxiety. Moreover, it is a novice to indicate the association between self-judgment and sleeping quality. Similar outcomes have been illustrated in the term "self-coldness" which is attributable to affecting sleeping quality (Houston et al., 2023). While, other research studies only indicated sleep quality as a covariate or moderator between self-compassion and physical health/health behaviors (Phillip and Hine, 2021; Pastore and Fortier, 2023). However, qualitative investigation might be needed to further explore the interplay between the variables. Despite the limited literature available, discussing the impact of physical activity on reducing psychological stress and sleep problems (Kaseva et al., 2019; Barham et al., 2022; Maddox et al., 2023), IGD was less likely to be investigated as predictors, so as in addition to the mediation role of self-compassion in potentially reducing the sleep problems impacted. Research studies on IGD and mental health have also indicated that resilience plays a beneficial role

between depressive symptoms and IGD and that mental health literacy enhances self-awareness and help-seeking behavior (Jeon et al., 2022).

Overall, the present study highlights the mechanism between IGD and sleeping quality. It also emphasizes the potential need for self-compassion and physical activity to reduce the impact caused by IGD, particularly its association with sleeping quality, depression and stress. The mechanism, therefore, supports the need for having interventions that focus on self-compassion and physical activity may improve psychological health, and facilitate individuals to better cope with stressors associated with IGD. In future studies, clinical interventions with self-compassion and physical activity counseling targeting individuals with IGD could be developed and investigated. Clinical interventions, using motivational interviewing, might be considered one of the clinical therapeutic techniques involving self-compassion and physical activity. In motivational interviewing, clients identify and address their uncertainty toward behavioral change. Using the guidance, empowerment and curiosity communication style, motivational interviewing promotes behavioral changes in addiction, including IGD (Zajac et al., 2020). Previous research studies evidenced that motivational interviewing was widely used in physical activity counseling and other community-based physical activity-promoting programs, and highlighted the long-term effect on behavioral changes, as well as individuals' awareness, attitude and beliefs toward an active lifestyle (Hardcastle and Hagger, 2011; Wade et al., 2020; Dziano et al., 2021). Additionally, the reframing, affirming, accepting and compassionate process within motivational interviewing demonstrates the link between motivational interviews and self-compassionate outcomes (Pastore and Fortier, 2020). Pastore *et al.* have also developed motivational interviewing-based physical activity counseling. Her research provided evidence supporting physical activity counseling in increasing physical activity behavior and self-compassion; nonetheless, individuals with increased self-compassion prioritize other health behaviors like sleep and healthy eating (Pastore et al., 2021). As a result, motivational interviewing could be considered a comprehensive technique to therapize IGD, while cultivating self-compassion and healthy lifestyle behaviors. It could be beneficial to combine self-compassion psychoeducation with active video gaming intervention and gradually transition to physical activity intervention, in addition to motivational interviewing. This approach aligns with the research that suggests that behavioral modification intervention alone is less effective and that active video games are an acceptable way to increase physical activity and reduce internet addiction. The integration of physical activity or active video gaming with psychoeducation and practical counseling healthcare services is anticipated to be more feasible and effective in real-world settings. This approach is preferable to attempting to control teenagers' behaviors or enforcing changes, as it is often challenging to maintain adherence over time. Apart from interventions and programs, further research, like longitudinal studies, is needed to better understand the mechanisms underlying the relationships between IGD, psychological outcomes, other possible moderators (*i.e.,* personality, coping style, parental style) and healthy lifestyle behaviors on a long-term basis.

**Open peer review.** To view the open peer review materials for this article, please visit http://doi.org/10.1017/gmh.2024.36.

**Data availability statement.** All data generated or analyzed during this study can be provided upon request.

**Author contribution.** G.Z.F.Y., C.L., and H.W.F collected the data for the study. W.M.Y.C and H.W.F conducted the data analysis and wrote the first draft of the manuscript with critical input from all authors. All authors reviewed the final draft of the manuscript.

**Financial support.** The research was not supported by any research grant.

**Competing interest.** The authors declare none

**Ethics statement.** This study obtained ethical approval at the Leshan Normal University, China (Ref: LNU-20230221R), and the study was conducted in accordance with the Declaration of Helsinki. Informed consent was obtained from all participants prior to their inclusion in the study.

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
