## [Editor Report]

Dear author, thank you very much for submitting the article and we apologize for the delay in the review. The reviewers have provided us with their suggestions and we are forwarding the same to you below. 

Best regards

1. First reviewer

This quantitative cross-sectional study analysed the relation between internet gaming disorder and sleep quality mediated by social relationship, depression, self-compassion, and physical activity. It is a novel study on a topic that greatly lacks attention from researchers. The manuscript is overall well written and pleasant to read. I noted some point that would benefit from further clarifications and some formulation that in my opinion should be reflected differently. I hope my comments are well received and considered as suggestions.

General comments: Please put the acronym IGD after the first appearance of internet gaming disorder in the summary and use it throughout your key practitioner messages.

Also revise your manuscript to use the IGD acronym throughout, or remove it from the first occurrence of internet gaming disorder.

I also notices some typos or grammatical errors throughout. A proof-reading could be useful.

Introduction: p.3 paragraph 1 line 18: attention to your vocabulary. Try replacing “normal” students by students without depression (or healthy, or something in that vein).

Method: Is there a rational for limiting the recruitment age to 24yo? If so please detail.

You say that it is a sub study from an international survey. Did you aim for specific countries? Please detail.

It isn’t clear if you used the PHQ-B or the PHQ-9 and why you only included 3 items of the PHQ-9. Please elaborate and clarify.

For the SCS, it isn’t clear whether the 10 items of the SCS-SF only cover the 3 dimensions you used (self-kindness, self-judgment, and mindfulness) or if you only used some of the items, or something else. Please clarify.

It seems there are words missing in your presentation of the SI-SQS. Please revise.

Discussion: I think it should be noted/explored that leisure time physical activity may replace internet gaming in the time spent. Therefore, individuals involved in more leisure time PA necessarily have less time to game and are therefore less at risk of having internet gaming disorder.

You cannot conclude of an impact of any of your variable on any other variable based on your study. being cross-sectional, you can only conclude there is a relation between those variables. Please rephrase your last paragraph accordingly. 

2. Second reviewer

The article investigates the relationships between Internet Gaming Disorder (IGD), sleep quality, self-compassion, physical activity, and psychological distress in young adults using a path analysis approach. The study found that physical activity plays a significant role in connecting these variables, emphasizing its potential to improve sleep quality and reduce psychological distress among individuals with IGD. Moreover, the research suggests that interventions focusing on promoting physical activity and developing self-compassion could be beneficial for individuals with IGD.

Comments:

Clarity and Structure: The article is well-structured and systematically outlines the research background, hypotheses, methods, and results. However, some sections could benefit from clearer articulation of the findings and a more straightforward presentation.

Introduction and Background: The background information is comprehensive, offering a clear overview of the significance of Internet Gaming Disorder, its prevalence, and its potential impact on mental health. However, there could be a concise summary at the beginning to outline the primary objectives and contributions of the study.

Literature Review: The article extensively reviews existing literature, which is essential for contextualizing the study. However, it might be beneficial to summarize key findings and focus on studies directly relevant to the proposed research questions for a more concise narrative.

Methodology: The methods section provides detailed information about the measures and data analysis procedures used. However, it might be helpful to explain the rationale behind the choice of specific assessment tools and analysis methods in relation to the research objectives.

Confirmatory Factor Analysis (CFA): The comprehensive analysis of measurement models demonstrated good internal consistency for most variables, ensuring the reliability of subsequent path analysis. However, the exclusion of the self-judgment variable due to poor fit is a limitation. It might be beneficial to explore alternative measurement strategies or evaluate other indicators that capture the intended construct of self-judgment for a more comprehensive understanding.

Correlation Matrix: The observed negative association between self-judgment and sleeping quality is a significant finding. However, a deeper exploration into the underlying reasons for this association could enhance the study’s validity. Further analysis or qualitative investigations might shed light on the nature of this relationship, providing richer insights into the interplay between self-judgment and sleep quality.

Path Analysis: The path analysis results showcasing the influence of physical activity in bolstering model fit is a commendable observation. However, the study’s dependency on physical activity as a linking variable warrants a thorough examination of potential confounders or moderators that could influence this relationship. Sensitivity analyses or subgroup analyses could strengthen the understanding of the role played by physical activity in the model.

Power Analysis: The use of a power analysis to justify the adequacy of the sample size is a methodological strength. However, providing additional details or justifications regarding the chosen effect sizes and alpha levels used in the power analysis would further solidify the rationale behind the sample size determination.

Discussion and Insights: The study’s elucidation of the complex relationships among IGD, mental health, and protective factors like self-compassion and physical activity is insightful. However, the discussion could benefit from a more nuanced exploration of potential alternative pathways or interaction effects between variables. Consideration of potential mediating or moderating factors that could further clarify the relationship dynamics would enhance the study’s contributions.

Implications and Future Directions: The proposed interventions focusing on self-compassion and physical activity counseling for individuals with IGD are promising. However, considering the multifaceted nature of mental health and behavioral interventions, exploring additional strategies or combining interventions might provide a more holistic approach. Moreover, emphasizing the practical implementation of these interventions in real-world settings would enhance the study’s applicability and relevance.

Grammatical suggestions to improve certain sentences:

Original Sentence: “The study found a strong correlation between internet gaming disorder and Facebook addiction.”

Revised Sentence: “The study identified a robust correlation between internet gaming disorder and addiction to Facebook.”

Original Sentence: “The study has shown that both IGD and general problematic internet use exhibit a strong negative correlation with adolescents’ quality of life.”

Revised Sentence: “The study demonstrated that both internet gaming disorder (IGD) and general problematic internet usage share a substantial negative correlation with the quality of life among adolescents.”

Original Sentence: “Despite there were only spreading literature reviewing the effect of physical activity in reducing psychological stress and sleep problems...”

Revised Sentence: “Despite the limited literature available, discussing the impact of physical activity on reducing psychological stress and sleep problems...”

Original Sentence: “This finding further supports the fact that physical activity promotes good sleep quality.”

Revised Sentence: “This finding further bolsters the evidence that physical activity fosters better sleep quality.”

Original Sentence: “The mechanism, therefore, supports the need for having interventions that focus on self-compassion and physical activity may improve psychological health...”

Revised Sentence: “Therefore, this mechanism underscores the necessity for interventions emphasizing self-compassion and physical activity, which can improve psychological health...”

Original Sentence: “In other words, it evidences self-compassion, which involves self-kindness and mindful awareness towards oneself, as a crucial protective factor against depression and anxiety.”

Revised Sentence: “In essence, it provides evidence that self-compassion, entailing self-kindness and mindful self-awareness, functions as a critical protective factor against depression and anxiety.”